# LLMs and the Abstraction and Reasoning Corpus: Successes, Failures, and the Importance of Object-based Representations

**Yudong Xu**                                                                *wil.xu@mail.utoronto.ca*
*Department of Mechanical & Industrial Engineering, University of Toronto*

**Wenhao Li**                                                                *chriswenhao.li@mail.utoronto.ca*
*Department of Mechanical & Industrial Engineering, University of Toronto*

**Pashootan Vaezipoor**                                                      *pashootan@cs.toronto.edu*
*Department of Computer Science, University of Toronto*
*Vector Institute for Artificial Intelligence*

**Scott Sanner**                                                             *ssanner@mie.utoronto.ca*
*Department of Mechanical & Industrial Engineering, University of Toronto*

**Elias B. Khalil**                                                          *khalil@mie.utoronto.ca*
*Department of Mechanical & Industrial Engineering, University of Toronto*
*Scale AI Research Chair in Data-Driven Algorithms for Modern Supply Chains*

**Reviewed on OpenReview:** *https://openreview.net/forum?id=E8m8oySvPJ*

## Abstract

Can a Large Language Model (LLM) solve simple abstract reasoning problems? We explore this broad question through a systematic analysis of GPT on the Abstraction and Reasoning Corpus (ARC) (Chollet, 2019), a representative benchmark of abstract reasoning ability from limited examples in which solutions require some "core knowledge" of concepts such as objects, goal states, counting, and basic geometry. GPT-4 solves only 13/50 of the most straightforward ARC tasks when using textual encodings for their two-dimensional input-output grids. Our failure analysis reveals that GPT-4's capacity to identify objects and reason about them is significantly influenced by the sequential nature of the text that represents an object within a text encoding of a task. To test this hypothesis, we design a new benchmark, the 1D-ARC, which consists of one-dimensional (array-like) tasks that are more conducive to GPT-based reasoning, and where it indeed performs better than on the (2D) ARC. To alleviate this issue, we propose an object-based representation that is obtained through an external tool, resulting in nearly doubling the performance on solved ARC tasks and near-perfect scores on the easier 1D-ARC. Although the state-of-the-art GPT-4 is unable to "reason" perfectly within non-language domains such as the 1D-ARC or a simple ARC subset, our study reveals that the use of object-based representations can significantly improve its reasoning ability. Visualizations, GPT logs, and data are available at `https://khalil-research.github.io/LLM4ARC`.

## 1 Introduction

It has been recently claimed that Large Language Models (LLMs) such as GPT-4 (OpenAI, 2023a) exhibit "sparks of artificial general intelligence" (Bubeck et al., 2023). As a result, the impressive question-answering and text generation abilities of pre-trained LLMs are already being deployed in rather consequential e-commerce and educational settings[1]. If LLMs are to be used to reliably solve complex, noisy, real-world

---

[1]`https://openai.com/blog/introducing-chatgpt-and-whisper-apis`

problems, one would expect them to be capable of reasoning in simple, unambiguous, idealized settings. By "reasoning", we here mean "using evidence, arguments, and logic to arrive at conclusions or make judgments", as defined in (Huang and Chang, 2022). While the performance of LLMs on arithmetic and language-based commonsense reasoning benchmarks has been the subject of recent analyses (see for example Section 4.1 of (Huang and Chang, 2022) for a brief survey), it is unclear whether LLMs exhibit the ability to generate abstract concepts based on a handful of "training" samples (Odouard and Mitchell, 2022).

To quantitatively measure the gap between machine and human learning, the Abstraction and Reasoning Corpus (ARC) was introduced in (Chollet, 2019). The author advocates for leveraging human-level intelligence as a frame of reference for evaluating general intelligence. To that end, he draws upon the work of developmental psychologists Spelke and Kinzler (2007) on the theory of Core Knowledge to determine axes along which human-like intelligence should be measured. Core Knowledge identifies four broad categories of innate assumptions that form the foundation of human cognition:

– **Objectness:** The ability to perceive the surroundings as consisting of cohesive, persistent, and non-interpenetrating objects.

– **Agentness and goal-directedness:** The tendency to perceive certain objects in the environment as intentional agents with goals, capable of contingent and reciprocal actions, while distinguishing them from inanimate objects.

– **Numerical knowledge:** Innate knowledge of abstract number representations for small numbers and the concepts of addition, subtraction and comparison between those numbers.

– **Elementary geometry and topology:** Knowledge of distance and basic 2D and 3D shapes.

The ARC, a benchmark of 1,000 image-based reasoning tasks, is thus proposed as a test of the above four core knowledge systems in humans or AI systems. Each task requires the production of an output image given a specific input, with 2 to 5 input-output image pairs provided as training instances to "learn" the underlying procedure (Figure 2). The training inputs are different from the actual test input, though they are solvable using the same (unspecified) procedure. Crucially, no acquired knowledge outside of the aforementioned priors is required to solve these tasks. Note that although these priors are explicitly described, the ARC tasks remain completely open-ended: objects can have different shapes and colors and form various relations with one another, and the grid size can also vary between tasks. This feature makes these problems not amenable to solving through search. This is in contrast to games like Go and Chess (Silver et al., 2018; 2017) where the search space is large but the set of moves is finite and fixed. In fact, so far the approaches that have employed a heuristic search via a predetermined set of transformations have all fallen short of generalizing to the hidden tasks; see Section 5.

Given the vast corpus of human knowledge that LLMs are typically trained on, one might wonder whether they could have acquired the priors listed above. To answer this question, we conducted a comprehensive study of GPT-3.5 and GPT-4 on the ARC. We contribute three high-level findings that we believe shed light on some intrinsic limitations of the LLM framework and, to some extent, how to resolve them:

1. **GPT fails on simple ARC tasks:** Using pure text encodings (Figure 3) of tasks such as those in Figure 2, GPT-4 can only solve 13/50 of the simplest ARC tasks (Section 2). We conduct a failure analysis which reveals that the culprit is the LLM's inability to maintain "object cohesion" across the lines of text that represent the ARC image grids.

2. **GPT does better when objects can be easily detected in text:** We hypothesize that the issue raised in point 1 is tied to the two-dimensional nature of the ARC grids. In Section 3, we introduce the 1D-ARC benchmark, a set of ARC-like tasks that can be represented as a single line of text. Relative to the 50 ARC tasks, GPT performs better on 1D-ARC but is far from perfect.

3. **An object-based representation boosts GPT's performance significantly:** Leveraging the object-centric graph abstractions of the ARC solver Abstract Reasoning with Graph Abstractions

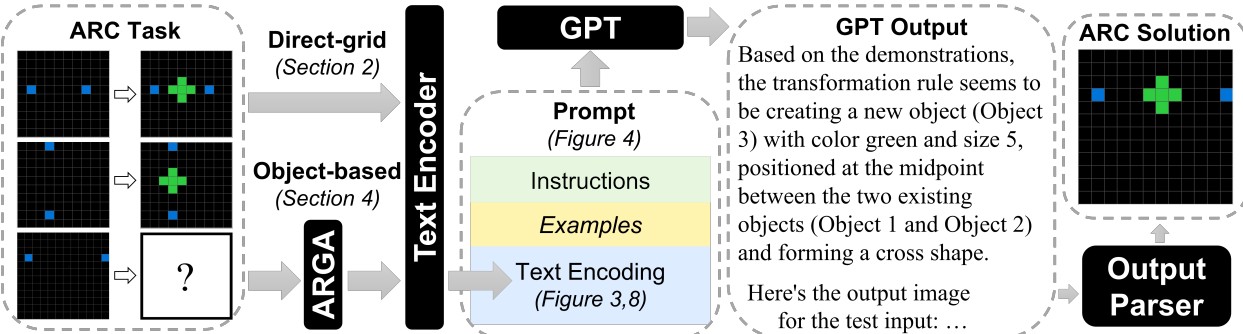

Figure 1: **Example of solving an ARC task with GPT.** An "ARC Task" consists of a set of training input-output pairs followed by a test input for which GPT should produce a correct output. To do so, a prompt is created. It includes high-level instructions on what GPT should do, optionally with additional in-context examples. A text encoding of the ARC task of interest is also included in the prompt. The encoding may be a direct representation of the 2-dimensional grids or an object representation produced by an external ARC solver, ARGA. GPT must then "reason" about the prompt to produce an answer. The output is then parsed and checked for correctness.

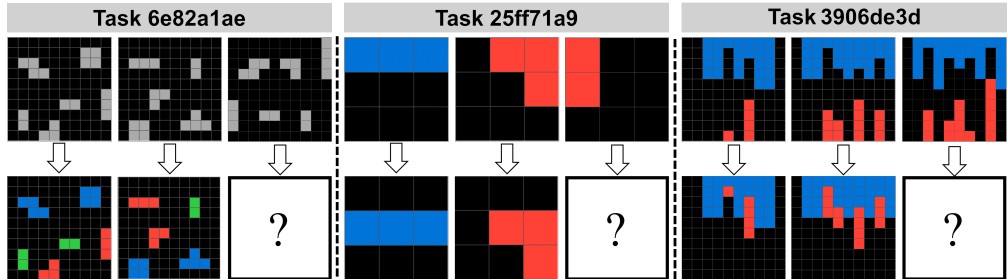

Figure 2: **Sample ARC Tasks.** Three tasks (separated by dashed lines) are shown. For a given task, each column contains one example input-output pair. The first two columns contain the "training" instances and the third column contains the "test" instance. The goal is to use the training instances to solve the test instance. The left task ("Recolor by size") requires recoloring the grey objects to green, red, or blue based on if their size is 2, 3, or 4. The middle task ("Static movement") requires moving the non-black object down 1 pixel. The right task ("Dynamic movement") requires moving the red objects up towards the blue objects until they make contact.

(ARGA) (Xu et al., 2023), we provide the LLMs with a more structured object-based representation of the input-output ARC grids (Section 4). This results in a significant jump in performance, where GPT-4 produces the correct output for 23 instead of 13/50 tasks and achieves near-perfect scores on many 1D-ARC task types. Further experiments on additional LLM models such as LLAMA-2 (Touvron et al., 2023) and datasets such as the full ARC training set and the Mini-ARC (Kim et al., 2022), although not the focus of this paper, exhibit a similar pattern.

Given the increasing interest from the artificial intelligence community in the reasoning capabilities of pre-trained LLMs and the unique characteristics of the ARC (and 1D-ARC), we believe that our work contributes to research on imbuing LLMs with such capabilities. We demonstrate that the use of an external tool that produces appropriate representations is crucial. We hope that our experimental design on the ARC, the new 1D-ARC dataset, and the integration of a domain-specific external tool for improved representation will be useful in generating new ideas at the intersection of LLMs and reasoning.

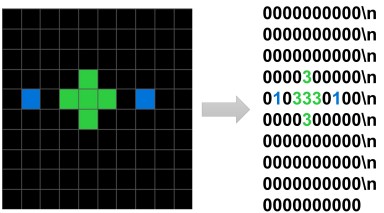

Figure 3: **Visualization of direct-grid encodings.** Left: each pixel is represented with a number corresponding to the pixel color, with no delimiters. Right: each pixel is represented with the color descriptor, separated by the delimiter "|". The text string has been formatted for easier reading.

## 2 A first attempt at solving ARC tasks with an LLM

The task of solving the ARC with an LLM necessitates the encoding of two-dimensional (2D) input-output images using a textual representation[2]. A text-encoded ARC task is incorporated into an LLM prompt, which then generates the solution. This section proposes a straightforward pipeline and evaluates it before mining the results to understand failure modes.

### 2.1 Textual encoding

A 2D grid with colored pixels can be directly encoded into text by representing each pixel's color either numerically (using values from 0 to 9, each representing one of the ten colors) or with color descriptors (e.g., "blue", "green", "black"). A delimiter delineates between adjacent pixels and "newline" characters were used to separate the rows in an image. We assessed the impact of different delimiters (",", "|", or no delimiter) on LLM performance. Figure 3 provides two visual examples of this *direct-grid encoding*.

### 2.2 Prompting and strategy

After encoding ARC images into text, we incorporate the latter into prompts that instruct the LLM to solve the task at hand. We explored two single-stage strategies for prompting the LLM.

**Few-shot learning:** By design, an ARC task is a few-shot learning task, providing a handful of training examples for generating a solution for a test example. Therefore, adopting the few-shot learning strategy is the most straightforward and intuitive initial approach for leveraging the inherent structure of ARC tasks. The prompt created using this strategy has two main sections: "instructions" and "task". The "instructions" section outlines the nature of an ARC task and the expected behavior of the LLM; it is the same across all tasks. The "task" section provides information about the single ARC task of interest, including its few-shot examples. This approach, in line with the classic few-shot learning concept, encourages the LLM to leverage the provided examples to solve the task. An example of this prompting strategy can be found in Figure 4.

**In-context few-shot learning with chain-of-thought:** Building on the few-shot learning strategy and drawing inspiration from chain-of-thought (CoT) prompting introduced in (Wei et al., 2022), we investigate a natural combination thereof. This approach enriches the learning context for the LLM by augmenting the original prompt with an "examples" section, which includes two simple ARC-like tasks—different from the actual task of interest—and their step-by-step solutions. This strategy not only leverages the inherent task examples but also provides a stable learning base of *in-context examples* for the LLM, encouraging a CoT response. As such, it assesses the LLM's capacity to generalize and apply knowledge acquired from a limited set of contextual examples to solve a similar task. An example prompt using this approach can be found in Figure 4.

---

[2]At the time of our research, GPT-4's vision API, which enables the model to accept image inputs, was not yet available. Consequently, this paper does not focus on the multimodal capabilities of GPT-4. However, we provide glimpses into these capabilities and their potential in Section 6 and Appendix G.

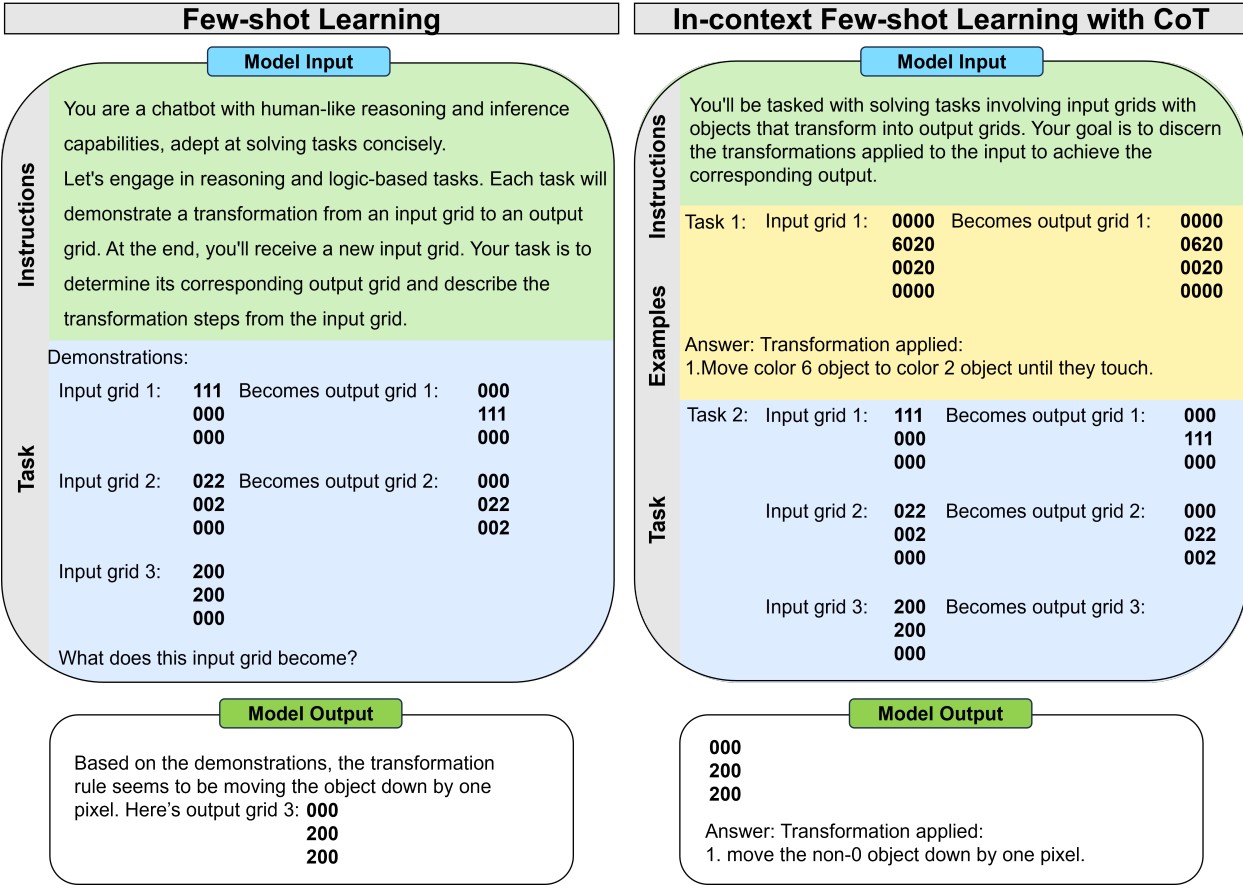

Figure 4: **Example prompts.** Left: Few-shot Learning. Right: In-context Few-shot Learning with CoT. The prompt texts have been formatted for easier reading (for example, the "Task" string on the left is provided to the LLM as "Demonstrations:\nInput grid 1: 111\n000\n000...").

Table 1: **Direct-grid variants, performance comparison.** Each row corresponds to a variant of a direct-grid encoding. Each column corresponds to a combination of a prompting method with either GPT-3.5 or GPT-4. GPT solutions were obtained through OpenAI's API with temperature set to 0. The values correspond to the number of tasks, out of 50, solved by each method; higher is better and top-performers are bolded.

| Direct-grid encoding | | Few-shot | | In-context Few-shot w/ CoT | |
| Pixel | Delimeter | GPT-3.5 | GPT-4 | GPT-3.5 | GPT-4 |
|---|---|---|---|---|---|
| Number | n/a | 3 | 5 | 2 | 9 |
| Number | \| | 4 | 11 | 5 | 12 |
| Word | , | 3 | 12 | 5 | **13** |
| Word | \| | 4 | 8 | 2 | **13** |

## 2.3 Results

Given that the most advanced ARC solvers achieve only a 30% accuracy on the hidden test set of the ARC, we strategically selected a subset of 50 ARC tasks. These tasks were among the "easiest", allowing our resources to be more efficiently allocated for in-depth experimentation. Our goal was to upper-bound the LLM's performance on the ARC as a whole; note that we do discuss results on all 400 ARC training tasks in the following paragraph and Section 4.2, but we restrict much of the analysis in this paper to the selected 50

tasks. We define "easy" tasks as those that have been previously addressed using the symbolic search-based method, ARGA (Xu et al., 2023). This designation stems from the fact that ARGA's implementation confines its solution space to 15 functions, in contrast to state-of-the-art models which can have a more expansive function set, with the Kaggle first-place solution, for example, encompassing 42 functions. Thus, studying this subset provides a clearer understanding of how a purely search-based solver, with a restricted function set, tackles ARC tasks.

The top-performing pixel representation and delimiter combination (Word + "|") solves only 13 out of the 50 tasks, as shown in Table 1. Using the top-performing representation, we assessed the complete ARC training set and GPT-4 managed to solve 81 out of the 400 tasks. Furthermore, on the Mini-ARC dataset—a simplified version of ARC detailed in Section 5—this approach solved 35 out of the 149 tasks. These accuracy rates are consistent with the performance observed on the subset of 50 tasks. In the following section, we will delve into the reasons why the LLM struggled with these tasks given that they are easy for a non-LLM method.

## 2.4 Analysis

We started our analysis by extracting key attributes such as pixel and color counts from each ARC task. We then applied logistic regression to explore potential relationships between these features and the performance of the LLM.

An intriguing finding from our analysis is that the number of colored pixels in a *test* image is associated with a notable negative coefficient, indicating a potential inverse relationship with the LLM's ability to solve tasks. Since a set of adjacent colored pixels often corresponds to an object in the ARC, this finding suggests that tasks with fewer objects are *more likely* to be solved by the LLM. Conversely, we find a positive coefficient associated with the average number of colored pixels in *training* images, implying a possible positive correlation with task solvability. This could suggest that more colored pixels in training images provide more learning material for the LLM, potentially improving performance. The full set of features studied can be found in Appendix E.

A closer examination of the tasks that GPT-4 solved correctly using the direct-grid approach reveals some interesting patterns in the reasoning provided by the model. Out of the 13 tasks that were correctly solved, only three tasks were accompanied by the correct reasoning steps. Surprisingly, for some tasks, GPT did not provide any reasoning at all, despite the presence of reasoning examples within the In-context Few-shot Learning with CoT prompts. This inconsistency in the application of reasoning illustrates a possible gap in GPT's understanding and application of the reasoning process, which further complicates the task of solving ARC problems. An example of a task where the reasoning provided by the model was incorrect despite achieving the correct output is illustrated in Figure 5. Further examples can be found in Appendix D.

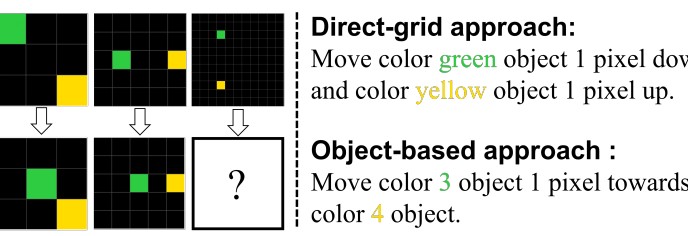

**Direct-grid approach:**
Move color green object 1 pixel down and color yellow object 1 pixel up.

**Object-based approach :**
Move color 3 object 1 pixel towards color 4 object.

Figure 5: **Reasoning provided by GPT-4 for an example task.** Both approaches produced the correct output grid. The direct-grid approach produced the wrong reasoning while the object-based approach produced the correct one.

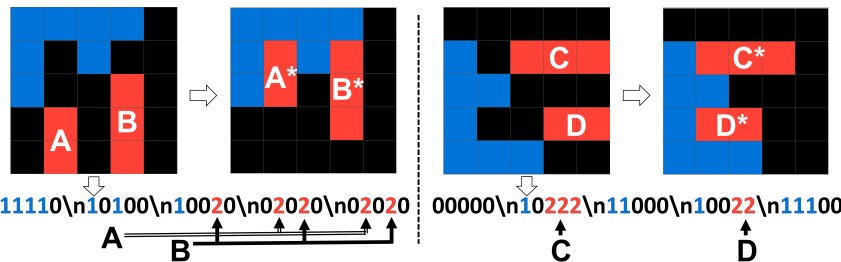

Figure 6: **Object cohesion analysis: two example tasks and their textual representations.** Generated based on the ARC task seen in Figure 2 (Right), the two tasks are identical modulo the 90-degree rotation. Left: objects A and B are vertical and become non-sequential when represented in text. Right: objects C and D are horizontal and become sequential when represented in text.

## 2.5 Object cohesion

To further understand the limitations of GPT on ARC tasks, we explored the concept of *object cohesion* in text, defined as the *"ability to parse grids into* 'objects' *based on continuity criteria including color continuity or spatial contiguity, and the ability to parse grids into zones, partitions"* (Chollet, 2019). Object cohesion is an integral part of human cognition (Spelke and Kinzler, 2007) and is assumed to be a significant part of the Core Knowledge priors required for ARC solving (Chollet, 2019).

Our objective was to investigate how the textual representation of objects influences GPT's problem-solving capacity. Given that the initial identification and abstraction of objects are pivotal in resolving the ARC (Acquaviva et al., 2022), understanding the impact of textual object depiction on the performance of language models is critical. We discovered that GPT's performance deteriorates significantly when objects are not sequentially represented within the text. To further demonstrate this, we selected tasks with clear horizontal or vertical objects and manipulated them to adopt the opposite orientation. A visualization of the difference between sequential and non-sequential object representation can be found in Figure 6.

For each "horizontal" or "vertical" original ARC task, we generated a rotated version and compared performance in both the horizontal and vertical configurations. An example is visualized in Figure 6 with more examples shown in Appendix C. The results in the leftmost part of Table 2 show a significant performance drop in the vertical case, reinforcing our hypothesis. It is evident that GPT struggles with object cohesion when objects are not sequentially arranged within the text. This insight not only deepens our understanding of the model's limitations but also guides us toward potential solutions. In the following section, we delve further into this finding by generating a new dataset that guarantees object sequentialness, and assessing the performance of LLMs on this dataset.

# 3 Does reduced task dimensionality improve LLM performance?

We introduce 1D-ARC, a novel variation on the original ARC that reduces its dimensionality to facilitate future research and provide a more approachable benchmark for LLMs. The 1D-ARC maintains the same Core Knowledge priors as the ARC but restricts the dimensionality of the input and output images to one dimension. Consequently, the images comprise only a single row of pixels, significantly reducing the complexity of tasks and enabling all objects to be represented within a single sequence. This modification effectively removes the challenge of maintaining object cohesion in non-sequential text. We visualize some example tasks in Figure 7 but the visualization for the full dataset is included in Appendix A.

The 1D-ARC dataset was strategically designed to adapt transformation types from the original ARC dataset to a one-dimensional format. This methodology effectively preserves the core knowledge priors inherent to the original ARC. An example of the design and generation process for a 1D-ARC task is illustrated in Appendix B.

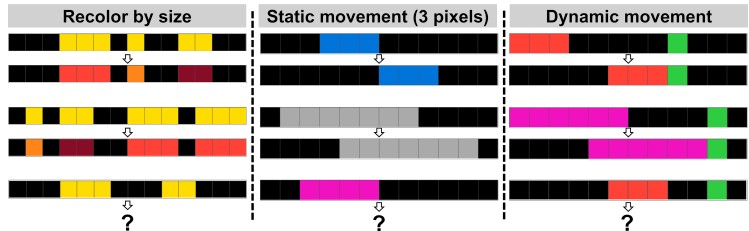

Figure 7: **Example tasks from the 1D-ARC dataset.** Each task is inspired by an ARC task; see Figure 2. From left to right: Recolor by size, Static movement by 3 pixels, Dynamic movement (move the block on the left until it touches the green pixel).

Table 2: **Results for direct-grid approach.** The number of solved tasks is out of 50. The first column is for the 50 tasks from the ARC. The second block of 5 columns is for some 1D-ARC task types; results on the full 1D-ARC can be found in Appendix A. The third block is for three task types (Fill, Move, Pile) with horizontal (H) and vertical (V) variants. All models were provided the same prompts for each task.

| LLM | ARC Subset | Move 1 Pixel | Move 3 Pixels | Move Dynamic | Recolor by Size | Denoise | Fill H | Fill V | Move H | Move V | Pile H | Pile V |
|---|---|---|---|---|---|---|---|---|---|---|---|---|
| GPT-3.5 | 2 | 10 | 7 | 6 | 2 | 13 | 2 | 0 | 0 | 0 | 2 | 0 |
| GPT-4 | 13 | 33 | 12 | 11 | 14 | 30 | 46 | 1 | 12 | 0 | 32 | 0 |
| LLAMA2-13B | 0 | 8 | 3 | 4 | 0 | 4 | 0 | 0 | 0 | 0 | 0 | 0 |
| LLAMA2-70B | 2 | 11 | 4 | 5 | 0 | 1 | 0 | 0 | 0 | 0 | 0 | 0 |

Our data generators have been developed to be capable of creating a variety of 1D-ARC tasks. They rely on task-specific parameters such as the maximum width of the 1D sequence, the maximum number of objects, and the maximum size of the objects. This parametric approach ensures tasks can originate from the same foundational concept or transformation yet manifest with varying complexities.

These generators can be seen as "1-concept" in design, each tailored for a specific transformation. However, their modular construction allows for the combination of multiple "1-concept" generators sequentially, leading to multi-concept tasks. As an example, a task could first necessitate the relocation of objects, then require a color transformation of these objects. This layered approach augments the scope and intricacy of the tasks, serving as a comprehensive evaluation of LLM capabilities.

### 3.1 Results

We used the best-performing prompts from Table 1 for the direct-grid approach and documented the results in the rightmost part of Table 2. Notably, the direct-grid encoding shows a relative improvement in performance on the 1D-ARC as compared to the original ARC. This implies that reducing both task space complexity and the spatial dimensionality of the input-output pairs enhances the LLM's ability to parse and reason with the encoded information.

Even with the significantly simpler 1D-ARC, there is still much room for improvement in performance. While GPT-4 is able to solve some tasks more effectively, it still falls short on others. This finding suggests that providing a sequential representation of objects in text alone may not be sufficient for GPT to effectively solve ARC tasks.

In light of these findings, we next explored the benefits of employing an external tool to perform object abstraction for GPT, thereby completely removing the challenge of object cohesion in text.

# 4    Enhancing LLM performance with an object-based representation

To address the challenges we have identified thus far and to enhance LLM performance, we propose the integration of an external tool to aid in producing an object representation of a task. More specifically, we leverage the ARGA algorithm (Xu et al., 2023) to execute object abstraction before prompting LLMs for the solution. We note that a task is deemed to be solved correctly when the LLM produces the correct textual representation of the output, given the input produced by the external tool. As such, whenever an LLM is coupled with ARGA, one should think of the integrated "ARGA+LLM" system as tackling the task, rather than the LLM alone.

## 4.1    Object-based textual representation

ARGA is a non-learning approach that aims to solve ARC by first abstracting the images into graph representations and then conducting a search within a Domain-Specific Language (DSL) defining possible changes to the graphs to identify the solution. We leverage the first component of ARGA to acquire a *graph representation* of the images. These graph representations, in which each node (or vertex) corresponds to an object in the image grid and each edge represents relationships between the objects, are subsequently encoded into object-oriented text representations. It is worth noting that ARGA provides a suite of hand-designed abstraction methods to cater to different tasks. In our case, we apply the "best-fit" abstraction, utilizing the abstraction that ARGA deems optimal for generating the solution for each task. Essentially, our objective is to evaluate the effectiveness of an abstraction mechanism that excels at object abstraction. After transforming the images into graphs, we examine two textual representations:

**Object Descriptors:**   This encoding technique presents a list of objects, each corresponding to a node in the graph and its associated attributes; see Figure 8 (Top). It offers a clear and intuitive representation of the image as a set of distinct objects, each carrying its own properties.

**Object JSON:**   On the other hand, the Object JSON encoding method provides a more structured representation of the graph; see Figure 8 (Bottom). This approach involves constructing a JSON list that encapsulates nodes and their corresponding attributes from the graph. The inherent organization of this format simplifies parsing and processing for the LLM, facilitating efficient extraction of pertinent information and relationships between the nodes.

Each encoding approach is further explored with an additional variant that includes edge information from the graph. In ARGA, edge information is utilized to identify relations between objects, as certain transformations applied to objects depend on other objects. For instance, an operation might involve recoloring an object to match the color of its neighbor. In the context of our study, our aim is to investigate whether the inclusion of edge information in the textual representation augments GPT's ability to solve ARC tasks.

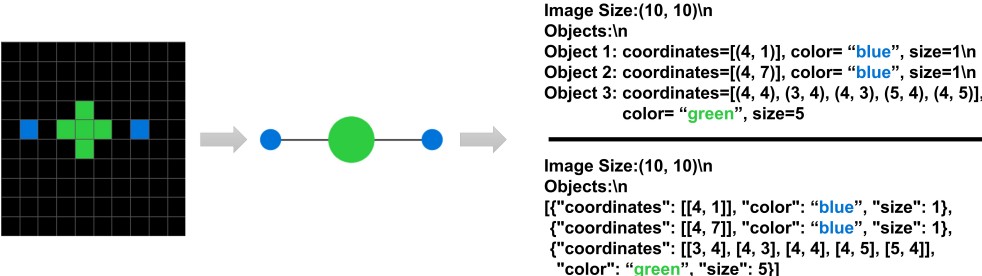

Figure 8: **Visualization of object-based textual encodings.** The 2D grid image is first transformed into a graph representation using ARGA. Then, the graph is encoded using the object descriptors representation (Top) or the object JSON representation (Bottom).

Table 3: **Object-based variants, performance comparison.** The values correspond to the number of tasks solved by each method, out of 50; higher is better and the top-performer is bolded.

| Object-based encoding | Few-shot | | In-context Few-shot w/ CoT | |
|---|---|---|---|---|
| | GPT-3.5 | GPT-4 | GPT-3.5 | GPT-4 |
| Object Descriptors | 9 | 16 | 5 | 20 |
| Object Descriptors with Edge | 5 | 18 | 4 | 12 |
| Object JSON | 8 | 21 | 6 | **23** |
| Object JSON with Edge | 5 | 19 | 4 | 16 |

Table 4: **Results for object-based approach.** In addition to the caption of Table 2, the numbers in parentheses are the increase in the percentage of tasks solved with the object-based approach compared to the direct-grid approach of Table 2. The number of tasks in each column is 50, so a 2% increase means that a given method has solved one more task.

| LLM | ARC Subset | Move 1 Pixel | Move 3 Pixels | Move Dynamic | Recolor by Size | Denoise | Fill | | Move | | Pile | |
|---|---|---|---|---|---|---|---|---|---|---|---|---|
| | | | | | | | H | V | H | V | H | V |
| GPT-3.5 | 6 (+8%) | 39 (+58%) | 14 (+14%) | 7 (+2%) | 21 (+38%) | 36 (+46%) | 17 (+30%) | 15 (+30%) | 1 (+2%) | 0 (-) | 7 (+10%) | 10 (+20%) |
| GPT-4 | 23 (+20%) | 50 (+34%) | 49 (+74%) | 37 (+52%) | 40 (+52%) | 50 (+20%) | 48 (+4%) | 49 (+96%) | 21 (+18%) | 20 (+40%) | 42 (+20%) | 37 (+74%) |
| LLAMA2-13B | 6 (+12%) | 29 (+42%) | 9 (+12%) | 7 (+6%) | 4 (+8%) | 17 (+26%) | 2 (+4%) | 3 (+6%) | 0 (-) | 0 (-) | 1 (+2%) | 0 (-) |
| LLAMA2-70B | 7 (+10%) | 39 (+56%) | 17 (+26%) | 6 (+2%) | 8 (+16%) | 19 (+36%) | 6 (+12%) | 5 (+10%) | 0 (-) | 0 (-) | 2 (+4%) | 0 (-) |

## 4.2 Results

We leveraged the prompting methods outlined in Section 2.2 in combination with our proposed object-based textual representations, replacing the direct-grid encoding. The results, presented in Table 3, show a marked improvement, with the success rate increasing from 13/50 tasks to 23/50 tasks on the ARC subset. Table 4 also shows that the previously observed performance gap between horizontal and vertical tasks in Table 2 is eliminated with the object abstraction, confirming our hypothesis that GPT's challenges with object cohesion in non-sequential text were the root cause. The orientation of objects becomes inconsequential, as desired. An even bigger performance boost is observed for the 1D-ARC, where GPT-4 achieves 50/50 on some task types. LLAMA-2 Touvron et al. (2023) models also exhibit an increase in performance for the object-based method.

On the complete ARC training set, the new object-based method allowed GPT-4 to solve 97 out of the 400 tasks, up from 81 previously. Likewise, on the Mini-ARC dataset, performance improved to 45 out of 149 tasks, up from 35. These results underscore the value of augmenting the LLM with an external tool that provides an appropriate representation, particularly when it comes to ARC tasks.

## 4.3 Analysis

We conducted the same solvability regression analysis from Section 2.4, observing the same correlations between task complexity attributes and solvability. Intriguingly, the models' performance was observed to decline when edge information was integrated into the representation. This unexpected result suggests that the influx of excessive information might overwhelm GPT, resulting in diminished performance. This discovery underscores the need for future research to find an optimal balance between supplying adequate contextual information and avoiding information overload.

Furthering our analysis, we performed a similar examination of the tasks correctly solved by GPT under the object-based approach. Out of the 23 tasks that produced the correct output, an impressive 20 tasks

exhibited correct reasoning (See Appendix D). This significantly improved reasoning performance underscores the impact of effective object abstraction on GPT's reasoning abilities. Figure 5 additionally showcases GPT's reasoning when prompted using the object-based approach for a task where the direct-grid approach initially fell short in providing accurate reasoning.

## 5    Related work

**Prompting methods for LLMs**    This is a very active area of development (Qiao et al., 2022; Huang and Chang, 2022). CoT prompting is introduced in (Wei et al., 2022), providing LLMs with intermediate reasoning steps leading to improved performance on some complex reasoning tasks. Extending this, (Kojima et al., 2022) demonstrated LLMs' potential as zero-shot reasoners by incorporating a "Let's think step by step" phrase in the prompt. This approach notably enhanced accuracy on various reasoning tasks, thus hinting at untapped zero-shot capabilities within LLMs that can be leveraged through simple prompting techniques.

**Augmented LLMs**    The survey by Mialon et al. (2023) emphasizes the potential of augmentation in overcoming limitations of a pure LLM approach. "Toolformer" self-learns to use external tools via APIs, significantly improving zero-shot performance across various tasks (Schick et al., 2023). Program-Aided Language models (PAL) (Gao et al., 2022) combines the strengths of LLMs with a Python interpreter to accurately solve some reasoning tasks.

**Solvers for the ARC**    Since the introduction of the ARC in 2019, various methods have been proposed to address it. A powerful DSL coupled with an efficient program synthesis algorithm has the potential to solve the ARC, as initially proposed in (Chollet, 2019). Notable examples include the Kaggle challenge's (Kaggle, 2020) winning solution, which utilized a manually-created DSL and DAG-based search for program synthesis (top quarks, 2020). Other high-ranking Kaggle participants followed similar strategies (de Miquel Bleier, 2020; Golubev, 2020; Liukis, 2020; Penrose, 2020). Fischer et al. (2020) employed a Grammatical Evolution algorithm within their chosen DSL, while (Alford et al., 2021) utilized the DreamCoder program synthesis system (Ellis et al., 2020) to derive abstractions from a basic DSL and compose solutions for new tasks through neural-guided synthesis. More recently, ARGA (Xu et al., 2023) was proposed as an object-centric framework that represents images using graphs and tree search for a correct program in a DSL based on the abstracted graph space. Alternative approaches for the ARC challenge have also been explored. The Neural Abstract Reasoner, a deep learning method, achieved success on a subset of ARC tasks (Kolev et al., 2020). Assouel et al. (2022) devised a compositional imagination technique to generate unseen tasks for enhanced generalization. (Ferré, 2021) focused on an approach based on descriptive grids. However, these alternatives have not yet surpassed state-of-the-art results.

**ARC-like datasets**    have been introduced to tackle the ARC's complexity. The Mini-ARC (Kim et al., 2022), a $5 \times 5$ compact version of the ARC, was generated manually to maintain the original's level of difficulty. The Sort-of-ARC (Assouel et al., 2022) shares ARC's input space but presents simpler problems with 20×20 images containing three distinct 3×3 objects. The ConceptARC dataset presents a set of manually crafted tasks, grouped and categorized by 16 distinct core concepts (Moskvichev et al., 2023). The PCFG benchmark (Mirchandani et al., 2023) is similar to our 1D-ARC and was generated using the probabilistic context-free grammar

**LLM for the ARC**    In a recent study (Moskvichev et al., 2023), the capabilities of both automated methods and human cognition were explored with respect to the ARC. Their research employed state-of-the-art ARC solvers (top quarks, 2020; de Miquel Bleier, 2020) and GPT-4 to tackle tasks originating from the ConceptARC dataset, comparing these solutions with those produced by humans. The approach to prompt GPT-4 was comparable to our few-shot direct-grid encoding method outlined in Section 2.2. This study revealed that GPT-4 lags significantly behind both the leading ARC solver and human performance, a finding that aligns with our own. Other recent studies also echo this finding (Mirchandani et al., 2023; Camposampiero et al., 2023). However, it is critical to highlight that, based on our investigations, the proficiency of GPT-4 on the ConceptARC dataset could potentially be enhanced by adopting an object-based representation in the prompting process.

While our paper focused on the effects of representation, a recent study has shown improvements in the performance of LLMs on the ARC by incorporating hypothesis search Wang et al. (2023). Additionally, language instructions from LARC (Acquaviva et al., 2022) were utilized to assist LLMs in solving the ARC, as demonstrated in another recent study (Pu and Alford, 2023). Although the improvement was notable, it was not exceptional, suggesting the presence of additional challenges which we discuss in Section 6.

## 6  Conclusion and Future work

We have explored the capabilities and limitations of the GPT LLM in solving ARC tasks seen as representatives of a certain kind of human-like intelligence. Our exploration started with a straightforward, grid-based textual encoding approach, which revealed that GPT struggles due to the non-sequential representation of complex objects in text. We then introduced the 1D-ARC, a simplified, single-dimensional version of the ARC. By reducing the task complexity and dimensionality, we aimed to make ARC tasks more approachable for LLMs. Our evaluations on the 1D-ARC indicated improvements in performance but also highlighted that simplification alone could not bridge all the gaps in GPT's reasoning processes. In the third phase of our exploration, we adopted an object-based approach, integrating an external tool, the ARGA framework, to assist in object abstraction. This led to significant improvements in GPT's problem-solving abilities, reaffirming the importance of structured, object-based representations in complex reasoning tasks.

Our research uncovers potential avenues for future exploration. For instance, edge information was not fully utilized by the LLM, suggesting that GPT-4 may not be capable of dealing with graphs when represented in text form. As we delve deeper into the possibilities of structured representations, we might consider introducing a "language" of transformations for LLMs to use in solving ARC tasks.

**Sub-abilities of LLMs**  Our research has predominantly focused on the LLM's ability to solve ARC tasks comprehensively. However, it's important to recognize that solving an ARC task can be a multi-step process, encompassing distinct sub-abilities. These include the LLM's capacity to abstract and understand objects from given inputs (Ability to Abstract), to formulate reasoning based on the understood input (Ability to Reason), and to generate accurate outputs using the reasoning and input (Ability to Execute). Each step presents unique challenges for LLMs. By this definition, our study can be viewed as assisting LLMs with the "Ability to Abstract". The recent study by Pu and Alford introduced in Section 5 can be seen as helping LLMs with the "Ability to Reason", since the reasoning is provided to LLMs directly. In both of these cases, LLMs still face difficulties in completely solving ARC tasks, which suggests that all three sub-abilities require further improvements.

Therefore, we believe that further experimentation aimed at dissecting and examining these sub-abilities could provide deeper insights into LLMs and pave the way for more sophisticated and capable language models in the future.

**Visual Inputs for LLMs**  Recently, advancements have been made in developing LLMs capable of processing image inputs. GPT-4V is one such example (OpenAI, 2023b). At the time of our research, the GPT-4V API was not available. We thus conducted experiments by manually prompting ChatGPT-4 with image attachments. Evaluating on the set of 50 tasks described in Section 2.3, we found that GPT-4V, at its current stage, is only able to solve 2 out of the 50 tasks; in contrast, GPT-4 solves 23 tasks and even LLAMA2-13B solves 6 tasks. An example of solving an ARC task with GPT-4V is depicted in Figure 9. More details about this experiment are provided in Appendix G. These findings demonstrate that GPT-4V with a multimodal visual and language input almost fails completely on the selected ARC tasks and significantly underperforms pure language-based LLMs, even pure language LLMs that are *much smaller* than GPT-4V. Hence, it appears there is room for significant improvement for multimodal LLMs to correctly reason about combined visual and language inputs in the context of ARC tasks, which should serve as a future challenge for LLMs like GPT-4V.

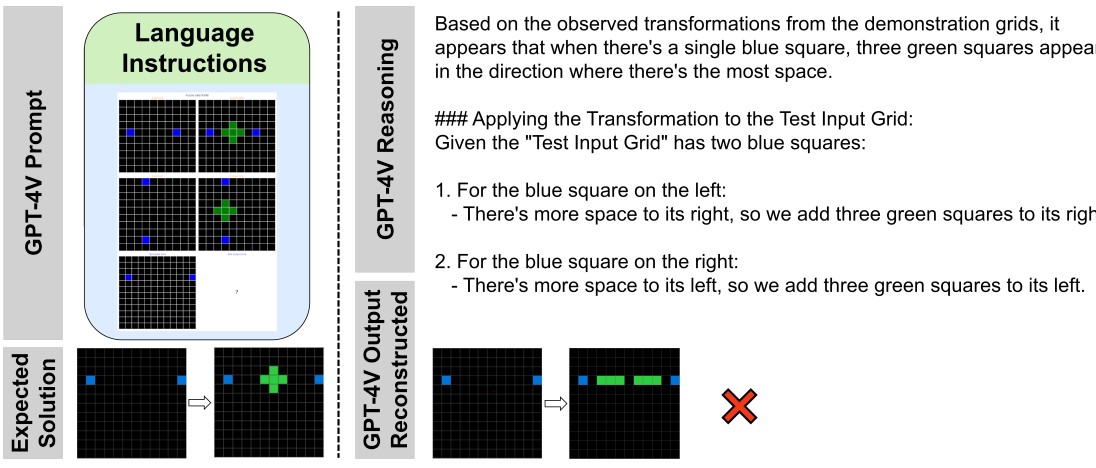

Figure 9: **GPT-4V solving an ARC task.** Language instructions as well as an image attachment were used as prompts to the model. GPT-4V describes the output in text. Subsequently, a corresponding grid was manually reconstructed to visually represent the text-based output described by GPT-4V.

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

## A 1D-ARC full dataset

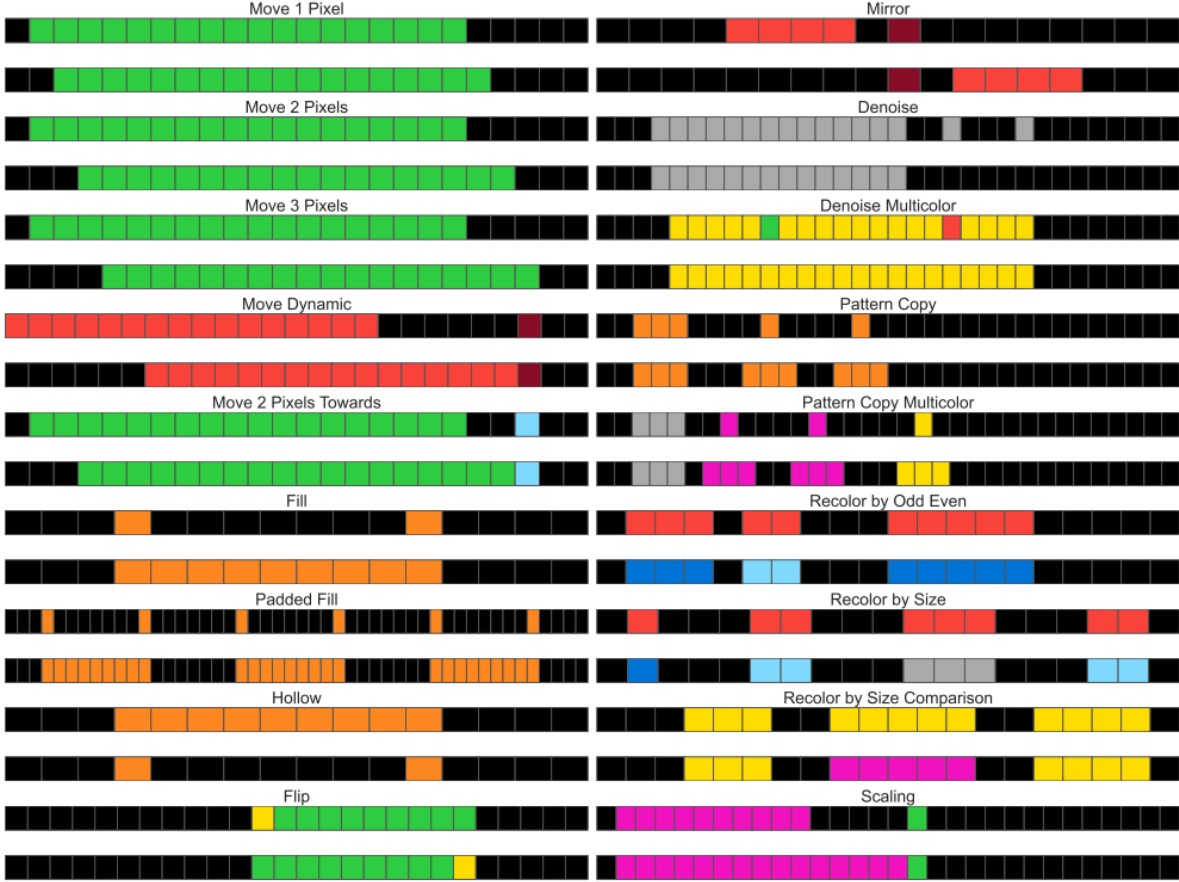

Figure 10: **1D ARC: visualizations of selected sample tasks.** For each task type, one pair of input-output images is shown.

Table 5: **Full results of direct-grid and object-based approach on the 1D-ARC dataset.** Each row displays results for one generated task type using different methods. The values correspond to the number of tasks out of 50; higher is better and the top-performer is bolded for each task type.

| Task | Direct-grid | | Object-based | |
|------|---------|--------|---------|--------|
| | GPT-3.5 | GPT-4 | GPT-3.5 | GPT-4 |
| Move 1 | 10 | 33 | 39 | **50** |
| Move 2 | 3 | 13 | 22 | **50** |
| Move 3 | 7 | 12 | 14 | **49** |
| Move Dynamic | 6 | 11 | 7 | **37** |
| Move 2 Towards | 3 | 17 | 17 | **50** |
| Fill | 6 | 33 | 44 | **49** |
| Padded Fill | 3 | 13 | 37 | **44** |
| Hollow | 2 | 28 | 40 | **48** |
| Flip | 11 | 35 | 20 | **50** |
| Mirror | 4 | 10 | 6 | **13** |
| Denoise | 11 | 18 | 48 | **48** |
| Denoise Multicolor | 13 | 30 | 36 | **50** |
| Pattern Copy | 11 | 18 | 31 | **45** |
| Pattern Copy Multicolor | 16 | 19 | 21 | **47** |
| Recolor by Odd Even | 13 | **16** | 15 | 13 |
| Recolor by Size | 2 | 14 | 21 | **40** |
| Recolor by Size Comparison | 6 | 10 | 17 | **28** |
| Scaling | 14 | 44 | 34 | **46** |

## B  1D-ARC Design Process

1. **Identify ARC task that requires core prior knowledge.**

   See Figure 11(Left).

2. **Determine the solution and the inherent knowledge priors.**

   **Solution:** Recolor objects based on their sizes.

   **Knowledge priors:**

   - Objectness (Object cohesion)
   - Numbers and Counting priors (Understanding of object size through counting)

3. **Develop the 1D-ARC task using the identified solution and knowledge priors.**

   See Figure 11(Right).

4. **Reiterate steps 1-3 to create a comprehensive 1D-ARC dataset.**

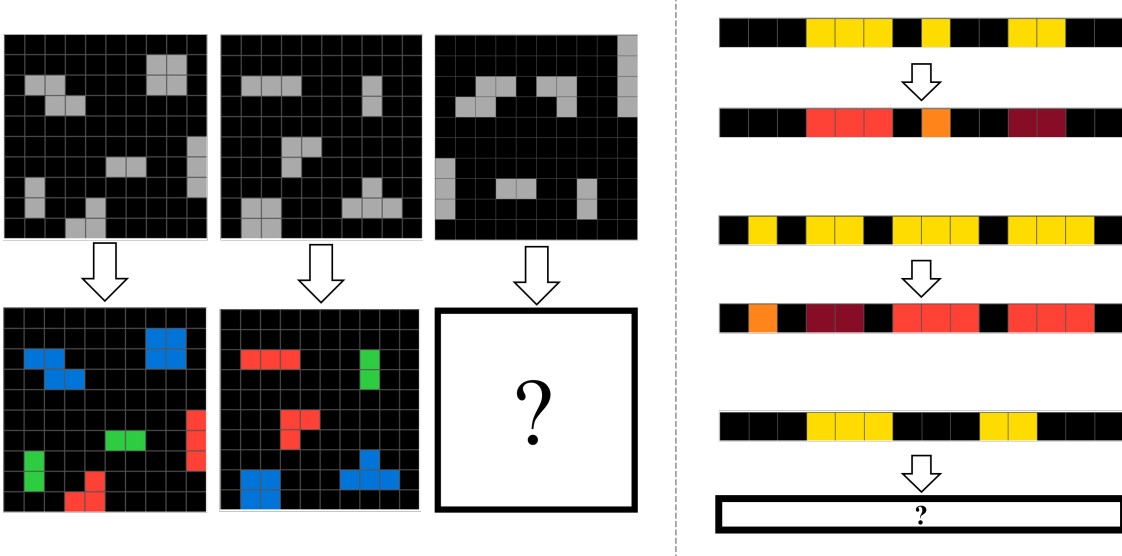

Figure 11: **Example of an ARC task (Left) and a 1D-ARC task adapted from it (Right)**

## C   Vertical and horizontal datasets

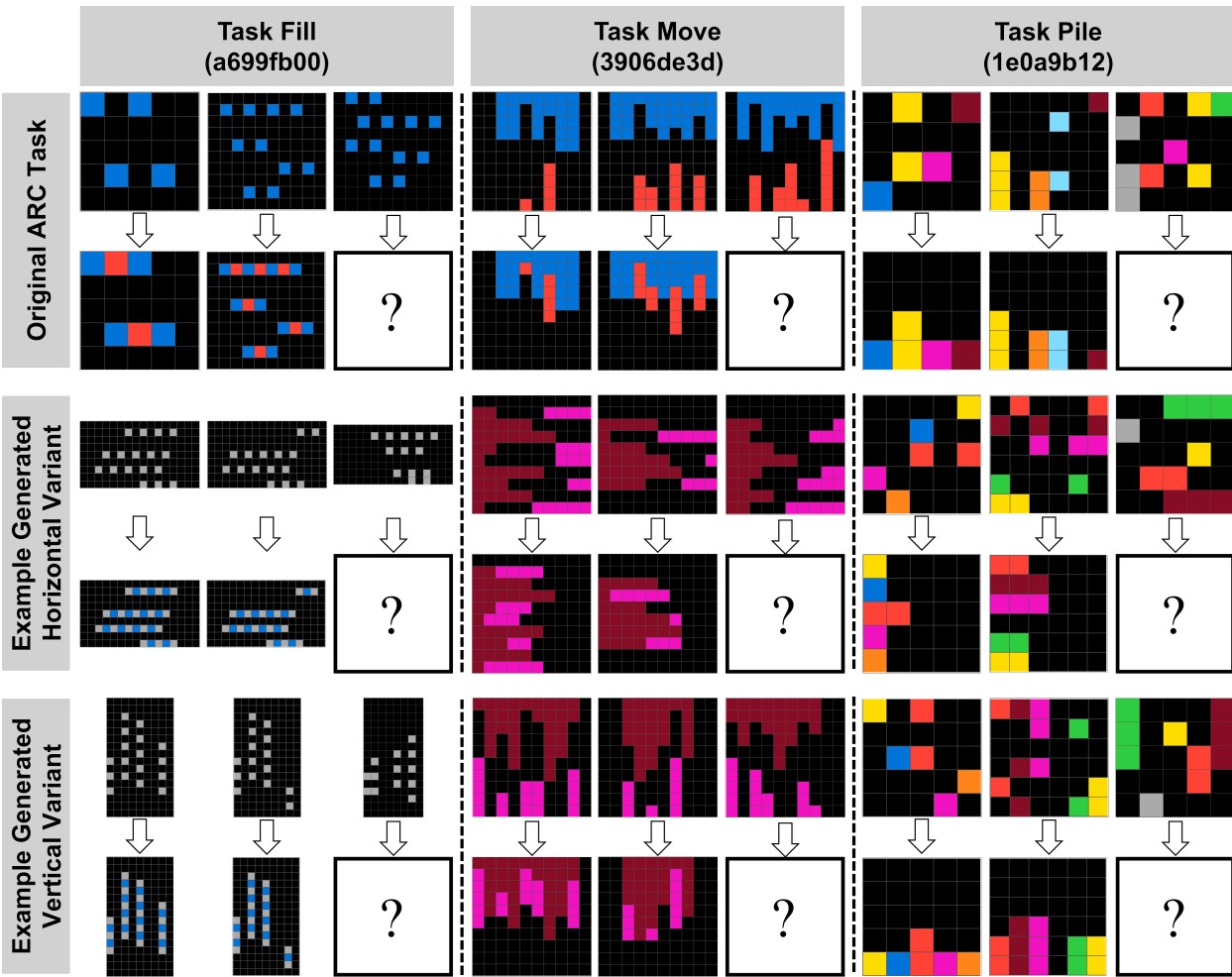

Figure 12: **Visualization for generated vertical and horizontal variants for three ARC tasks.** Each column displays the original ARC task on the top, one generated horizontal variant in the middle, and one generated vertical variant at the bottom.

## D   GPT reasonings for solved tasks

Table 6: **GPT Reasoning for correctly solved ARC tasks using best performing direct-grid approach.** The correctness of the reasoning provided is determined manually. If a reasoning was not provided, it is reported as N/A.

| Task | Reasoning | Correctness |
|------|-----------|:-----------:|
| 25ff71a9 | Transformation applied:
1. Move color objects (blue or red) 1 row down. | ✓ |
| aabf363d | N/A | ✗ |

| Task | Reasoning | Correctness |
|------|-----------|-------------|
| d5d6de2d | Transformation applied:
1. Replace color red object with color green object
2. Remove the outer border of the color green object | ✓ |
| dc433765 | Transformation applied:
1. Move color green object 1 pixel down and
   color yellow object 1 pixel up. | ✗ |
| b1948b0a | N/A | ✗ |
| 74dd1130 | N/A | ✗ |
| c0f76784 | Transformation applied:
1. Fill the empty spaces inside the gray objects with the following colors:
- Cyan for the top object
- Orange for the middle object
- Purple for the bottom object | ✗ |
| 6150a2bd | N/A | ✗ |
| c8f0f002 | N/A | ✗ |
| 67385a82 | N/A | ✗ |
| 67a3c6ac | N/A | ✗ |
| a5313dff | Transformation applied:
1. Replace the black cells inside the red shape with blue cells. | ✓ |
| a699fb00 | Transformation applied:
1. Add a red object to the right of every blue object,
except when there is already a red object to the right of the blue object. | ✗ |

Table 7: **GPT Reasoning for correctly solved ARC tasks using best performing object-based approach.** The correctness of the reasoning provided is determined manually. If a reasoning was not provided, it is reported as N/A.

| Task | Reasoning | Correctness |
|------|-----------|-------------|
| a79310a0 | Transformation applied:
1. Move color 8 object 1 pixel to the right
2. Recolor color 8 object to color 2 | ✓ |
| 25ff71a9 | Transformation applied:
1. Move the object 1 pixel to the right. | ✓ |
| aabf363d | Transformation applied:
1. Recolor the larger object with the color of the smaller object. | ✓ |
| d5d6de2d | Transformation applied:
1. Remove color-2 objects
2. Recolor color-0 objects to color-3 | ✓ |
| 6c434453 | Transformation applied:
1. For each color-1 object with size 8,
   remove the corners to create two color-2 objects with size 5. | ✓ |

| | | |
|---|---|---|
| dc1df850 | Transformation applied:
1. Surround each color-2 object with color-1 objects in a 3x3 square,
  excluding the center where the color-2 object is located. | ✓ |
| aedd82e4 | Transformation applied:
1. Recolor all color-2 objects that are not touching the
  largest color-2 object to color-1. | ✓ |
| dc433765 | Transformation applied:
1. Move color 3 object 1 pixel towards color 4 object. | ✓ |
| 6e82a1ae | Transformation applied:
1. Recolor color 5 objects to color 1 if size is 4
2. Recolor color 5 objects to color 2 if size is 3
3. Recolor color 5 objects to color 3 if size is 2 | ✓ |
| b1948b0a | Transformation applied:
1.Recolor color 6 objects to color 2 | ✓ |
| 4258a5f9 | Transformation applied:
1. For each color-5 object, create a color-1 object
  with 8 pixels surrounding the color-5 object. | ✓ |
| ded97339 | Transformation applied:
1. Connect color-8 objects with the same color in a straight line. | ✓ |
| 7f4411dc | Transformation applied:
1.Remove all color objects with size 1. | ✓ |
| e9614598 | Transformation applied:
1.Find the midpoint between the two color-1 objects.
2.Create a color-3 object of size 5 in a cross shape
  centered at the midpoint. | ✓ |
| ea32f347 | Transformation applied:
1. Recolor the largest color-5 object to color-1
2. Recolor the smallest color-5 object to color-2
3. Recolor the remaining color-5 object to color-4 | ✓ |
| 6150a2bd | Transformation applied:
1.Rotate the colors of the objects 90 degrees clockwise. | ✗ |
| 913fb3ed | Transformation applied:
1. For each object, create a new object with
  8 coordinates surrounding the original object.
2. Assign colors to the new objects based on the original object's color:
  color 2 -> color 1, color 3 -> color 6, color 8 -> color 4. | ✓ |
| c8f0f002 | Transformation applied:
1.Recolor color 7 objects to color 5 | ✓ |
| 67385a82 | Transformation applied:
1.Recolor color 3 objects with size > 1 to color 8. | ✓ |
| ce22a75a | Transformation applied:
1. Surround each color-5 object with color-1 objects in a 3x3 square,
  keeping the color-5 object in the center. | ✗ |

| | | |
|---|---|---|
| 67a3c6ac | Transformation applied:
1.Rotate the colors clockwise within the objects. | ✗ |
| a5313dff | Transformation applied:
1.Recolor color 0 objects to color 1 | ✓ |
| a699fb00 | Transformation applied:
1. Recolor color 0 objects to color 2 | ✓ |

# E  Logistic regression analysis

| Feature | Coefficient | |
|---|---|---|
| | Direct-grid | Object-based |
| Number of colored pixels in test input image | -0.151312 | -0.365261 |
| Average number of colored pixels in training input images | 0.215891 | 0.326572 |
| Number of unique colors in test input image | -0.282226 | 0.346230 |
| Average number of unique colors in training input images | 0.192485 | -1.186780 |
| Number of pixels changed in test instance | 0.110529 | 0.142800 |
| Average number of pixels changed in training instances | -0.152656 | -0.090327 |
| Test input image size | -0.004665 | 0.001771 |
| Training input images average size | -0.013070 | -0.005959 |
| Number of training instances | 0.297392 | 0.158643 |

| Performance Metric | Score | |
|---|---|---|
| | Direct-grid | Object-based |
| Precision (unsolved) | 0.78 | 0.83 |
| Precision (solved) | 0.44 | 0.73 |
| Recall (unsolved) | 0.86 | 0.74 |
| Recall (solved) | 0.31 | 0.83 |

Table 8: **Results of logistic regression analysis.** Top: Comparison of feature coefficients for the best-performing direct-grid and object-based approaches, demonstrating the impact of each feature on an ARC task's solvability. Bottom: Precision and recall scores of logistic regression model for solved and unsolved tasks.

## F   Visualization of GPT solutions on example ARC tasks

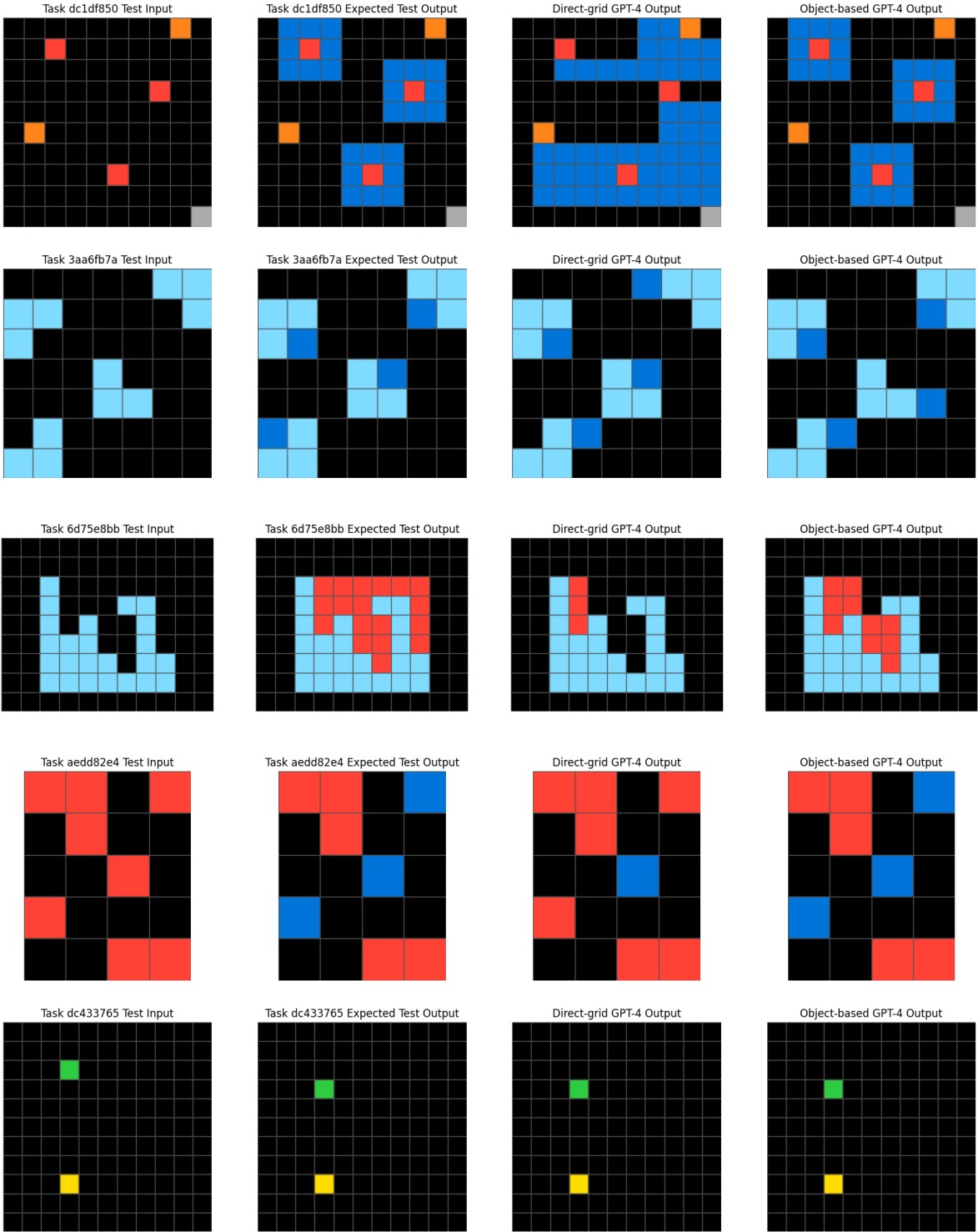

Figure 13: **Visualization of example ARC Tasks.** Each row showcases an ARC task. The first column displays the test input for that task, while the second column shows the expected test output. The third and fourth columns present the predicted outputs using the best-performing direct-grid approach and object-based approach, respectively.

# G   Experiments with GPT-4V

This section briefly introduces the preliminary experiments conducted for GPT-4V. The GPT-4V API was not available at the time of our research, therefore, the experiments were conducted by manually prompting ChatGPT-4 with image attachments. First, the prompting method is described and visualized, followed by a display of the results. Examples of GPT's output will also be shown.

## G.1   Prompting

The prompting style follows the few-shot approach introduced in Section 2.2. It consists of "instructions" and "task" sections. The "instructions" section is similar to before where it outlines the nature of an ARC task and the expected behavior of the LLM. The "task" section provides information about the ARC task of interest, however, instead of using textual representations, an image containing the few-shot examples as well as the test input is provided. Figure 14 shows an example of the prompts.

## G.2   Results

GPT-4V cannot currently produce images as outputs, so it produces texts that describe the output grid. To evaluate the results, we manually examined GPT-V4's outputs and reconstructed the output images based on the descriptions it produced. We found that GPT-4V was only able to correctly describe the output image for 2 tasks out of the 50 tasks we defined earlier, even more so, only a further 3 out of 50 tasks had the correct logic before producing the wrong output. 2 examples of ARC tasks as well as GPT-4V outputs for them can be seen in Figure 15

Figure 14: **Visualization for prompting GPT-4V with an ARC task.**

## Task e9614598

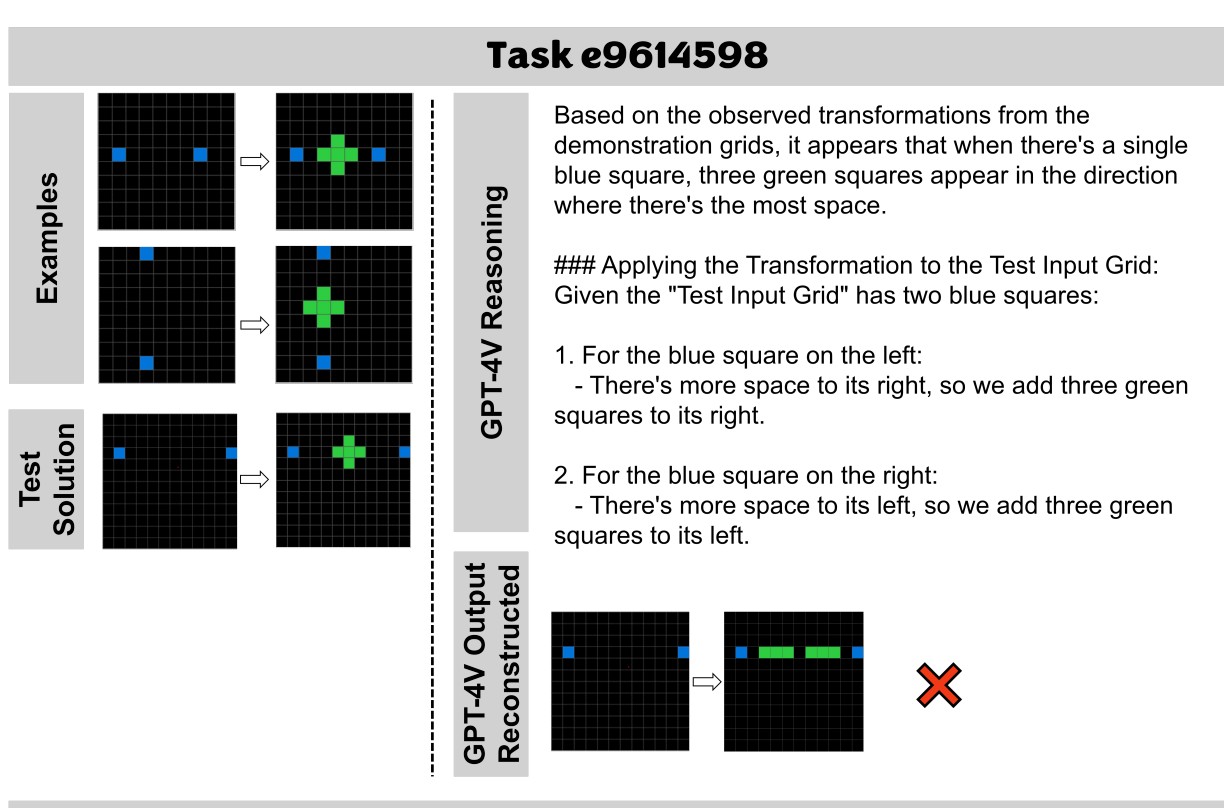

**GPT-4V Reasoning**

Based on the observed transformations from the demonstration grids, it appears that when there's a single blue square, three green squares appear in the direction where there's the most space.

### Applying the Transformation to the Test Input Grid:
Given the "Test Input Grid" has two blue squares:

1. For the blue square on the left:
   - There's more space to its right, so we add three green squares to its right.

2. For the blue square on the right:
   - There's more space to its left, so we add three green squares to its left.

**GPT-4V Output Reconstructed**

## Task 913fb3ed

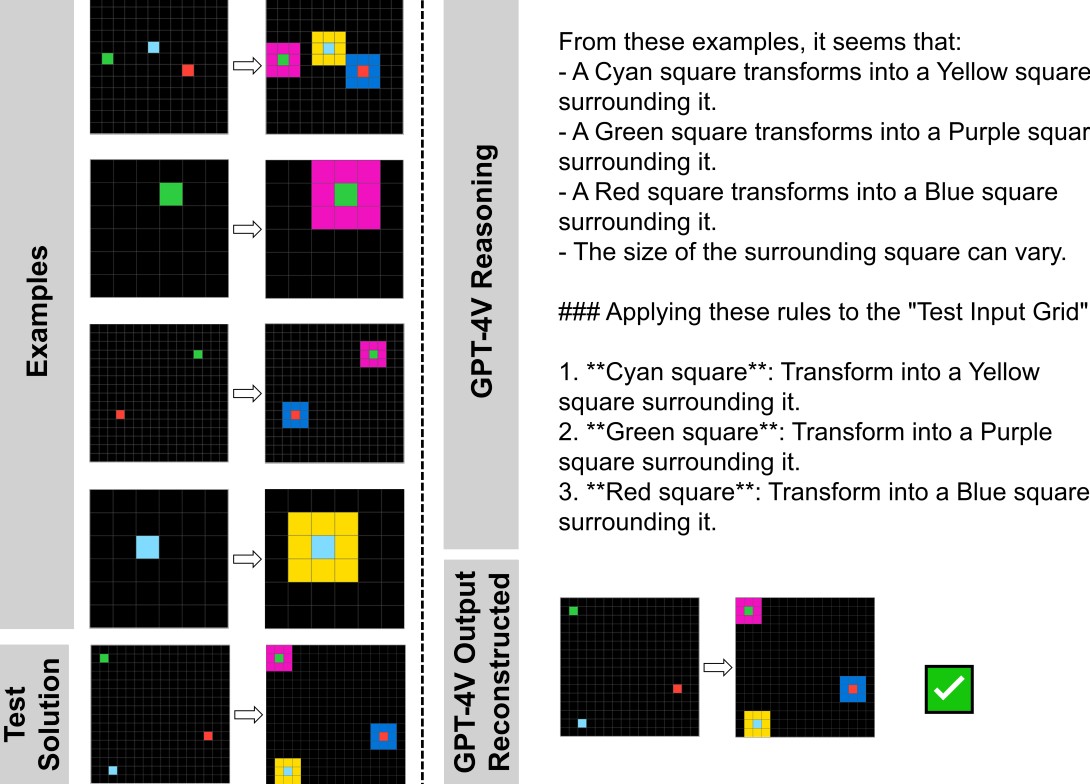

**GPT-4V Reasoning**

From these examples, it seems that:
- A Cyan square transforms into a Yellow square surrounding it.
- A Green square transforms into a Purple square surrounding it.
- A Red square transforms into a Blue square surrounding it.
- The size of the surrounding square can vary.

### Applying these rules to the "Test Input Grid":

1. **Cyan square**: Transform into a Yellow square surrounding it.
2. **Green square**: Transform into a Purple square surrounding it.
3. **Red square**: Transform into a Blue square surrounding it.

**GPT-4V Output Reconstructed**

Figure 15: **Visualization of GPT-4V outputs for 2 ARC tasks.**

