# OpenReview forum: "LLMs and the Abstraction and Reasoning Corpus: Successes, Failures, and the Importance of Object-based Representations"
_TMLR — Accepted by TMLR_

### Review · Reviewer_uAxr · 2023-12-04

**Summary Of Contributions:**

This article offers an analysis of the poor abstract reasoning
capabilities demonstrated by the widely acclaimed large language
models (LLMs) GPT-3.5 and GPT-4, based on the existing set of
benchmarks called ARC (stands for Abstraction and Reasoning Corpus).

The article claims that the lack of the LLMs' reasoning power for the
ARC puzzles stems from their lack of spacial understanding of these
puzzles. Namely, an ARC puzzles represent a 2-d grid demonstrating
colored shapes while LLMs are designed to generate text in the context
of 1-d streams of information. This basically results in the model's
failing to understand that some of the bits of information are related
to each other and represent the same object in the grid. The authors
propose a test confirming their hypothesis, which consists in
transposing the stream and showing the performance of both versions of
GPT drops down even more on the transposed benchmarks.

Furthermore, the article proposes a simplified set of benchmarks
called 1D-ARC, where all the puzzles are described by a 1-d image, to
check LLMs' abstract reasoning capabilities on benchmarks where the
data is stream-like. The authors show that both versions of GPT
exhibit better performance on this set of benchmarks as the models do
not have to reason about 2-d information.

Finally, the authors adapt the existing technology for translating the
ARC grids into a DSL-like symbolic representation of objects appearing
in the puzzles, which helps GPT-* models further increase their
performance.

**Audience:**

Yes

**Claims And Evidence:**

Yes

**Requested Changes:**

Please incorporate the use of GPT-4V in the text, the experimental
result and their analysis.

**Strengths And Weaknesses:**

In terms of pros, I should first say that although I am no expert in
the field of generative AI, I find the paper to be pretty easy to
follow. There are no technicalities that would represent a challenge
for an average reader. The topic and the discussion are quite simple
to understand. Second, the submission is accompanied by an archive
containing the experimental data, which is also a plus. Finally, the
conclusions drawn by the authors make perfect sense to me.

As for cons, I have to say they result from (some of) the same aspects
I identified as pros. Namely, the insights offered by the manuscript
are somewhat shallow, which affects the observations and conclusions
made. I would say the submission lacks some "essence juice" that would
make me think it is valuable. Namely, it is really unsurprising that
LLMs have a hard time with 2-d data as they are designed to predict
the next token in a one-dimensional context *sequence*. And it seems
plain that a rigorous textual/symbolic summary of a puzzle, which the
authors apply as the final ammunition, should provide an LLM with
higher chances of success in solving the puzzle.

In general, the trend of looking for magical (reasoning) properties of
generative AI and applying it all over the place for the tasks it does
not fit looks weird to me. Granted I come from a different area in AI,
I will not say this particular criticism of mine justifies rejection.
The work seems to have merits: it studies quite a fun problem and
applies uncomplicated but sensible mechanisms to analyze and address
it.

Having said that, I find it strange that the authors say they
performed only *preliminary* experimentation with GPT-4V despite the
existence of the API. And I feel the article will definitely benefit
from a full-blown comparison of the upsides and the downsides of the
proposed ideas versus GPT-4V. Otherwise, the findings look incomplete
to me.

---

> ### Author Response · Authors · 2023-12-17
>
> Thank you for your feedback and criticisms.
>
> **Significance of findings**
>
> We agree that our conclusion—that object-based prompting improves LLM performance—may seem intuitive. However, this very intuitiveness underscores the importance of our work. Despite its straightforward appeal, the object-based approach to evaluating the ARC is not a common practice.
>
> To the best of our knowledge, our research is novel in implementing and demonstrating the effectiveness of this approach, a contrast to the prevalent direct-grid methods in other studies. By introducing and validating an innovative prompting approach, we believe our study offers a fresh perspective and paves the way for more diverse applications in the realm of LLMs.
>
> **GPT-4V results**
>
> We agree with your point regarding the comparison of GPT-4V with the models discussed in our paper. In response to your suggestion, we will relocate the results of our GPT-4V experiments from the appendix to the main body of the article to provide a more direct and comprehensive comparison.
>
> Regarding the preliminary nature of our results with GPT-4V, it's important to note that the GPT-4V API was not accessible when we were conducting the research. Consequently, our experiments were limited to manual interactions with the ChatGPT's GPT-4 model, which accepts vision prompts.
>
> However, our findings indicate that GPT-4V's performance is significantly inferior to the other approaches we have tested. This led us to prioritize exploring other aspects of LLMs that could potentially enhance performance. We plan to revisit GPT-4V in future research, as its capabilities evolve and improve.

---

> > ### Comment · Reviewer_uAxr · 2023-12-20
> > **Reply to authors**
> >
> > Thanks a lot for your response. I look forward to a more comprehensive comparison with GPT-4V!

---

### Review · Reviewer_RCQt · 2023-12-06

**Summary Of Contributions:**

This paper looks at the abstraction and reasoning capability of large language models, largely based on the Abstraction and Reasoning Corpus (ARC) and the author-proposed derivatives. The benchmarks require the model to identify patterns among input-output pairs, which are specified as textual characters. The authors conducted a suite of investigations, studying the effect of various factors, such as "object" representation, input dimensionality, and few-shot prompting. Significant gaps have been observed between GPT performance and ideal performance, and the object representation has been found to be most influential in ablation studies.

**Audience:**

Yes

**Broader Impact Concerns:**

No apparent ethical implications.

**Claims And Evidence:**

Yes

**Requested Changes:**

See the weaknesses section.

**Strengths And Weaknesses:**

Srengths:

1. The paper is clearly written, with helpful figures illustrating the task setups.

2. The investigation is well-motivated, and I particularly like the use of the heuristic parsing algorithm.

Weaknesses:

1. Only two closed-source models have been studied. Could any of the recent open-source models be used, such as LLAMA 2?

2. Using open source models could also enable various attention visualizations, which may shed light on the observed findings.

3. A baseline of how well the model understands the object representation should be conducted. In this baseline, the ground truth reasoning pattern is given. If the model performs well in generating the new output according to the pattern, then it suggests an issue of pattern finding. However, if the model still couldn't perform well, then it seems to suggest that the model is having trouble understanding the textual input as a "grid".

4. The benchmark is by definition quite synthetic. I am wondering if there could be real-world extensions/adaptations that highlight the practical use of such reasoning ability.

---

> ### Author Response · Authors · 2023-12-17
>
> Thank you for your review and suggestions.
>
> **Further evaluation using open-sourced models**
>
> As per the global response, we are currently extending our research to include Llama 2 models and will report the results in a subsequent response as well as revise the paper.
>
> **Baseline experimentations**
>
> We appreciate the suggestion of conducting a baseline experiment, however, it is difficult to conduct such experiments since a singular 'ground truth' reasoning pattern is often non-existent. Solutions can be articulated in multiple ways using natural language, and there may be various valid solutions for a given task. Nonetheless, Acquaviva et al. have made strides in this area with their 'Language-complete Abstraction and Reasoning Corpus (LARC)' [1], which offers reasoning in natural language for each task. Preliminary experiments conducted by Pu et al [2] presented GPT-4 with language instructions from LARC and input akin to our direct-grid approach, resulting in the model successfully solving 104 out of 400 tasks in the training dataset.
>
> This outcome signifies a notable improvement compared to scenarios where no logical reasoning is provided. This indicates that a key challenge for LLMs is deciphering the logical reasoning behind tasks. However, the improvement is notable but not exceptional which points to additional potential challenges. These could include GPT-4's potential limitations in grasping complex reasoning even when given, the clarity of human-provided descriptions, or difficulties in correctly identifying objects and their relationships from the input, etc. While further exploration of these aspects is valuable, it falls beyond the scope of our current paper, which is primarily focused on the impact of representation alone. Future work could delve deeper into these issues to gain a more comprehensive understanding. We will include this future work discussion on revision.
>
> **Potential real-world extensions**
>
> The ARC might appear simplistic and disconnected from real-world applications. However, it is designed as a benchmark for AI and assumes similar core knowledge priors innate in humans [3]. The ability to form and abstract such concepts is central to human intelligence [4], and thus, it is essential for these capabilities to be replicated and processed by AI systems.
>
> [1] Acquaviva, Sam, et al. "Communicating natural programs to humans and machines." Advances in Neural Information Processing Systems 35 (2022): 3731-3743.
>
> [2] Yewen Pu, et al. “larc solving with gpt4” https://github.com/evanthebouncy/larc_gpt4 (2023)
>
> [3] Chollet, François. "On the measure of intelligence." arXiv preprint arXiv:1911.01547 (2019).
>
> [4] Moskvichev, Arseny, Victor Vikram Odouard, and Melanie Mitchell. "The ConceptARC Benchmark: Evaluating Understanding and Generalization in the ARC Domain." arXiv preprint arXiv:2305.07141 (2023).

---

### Review · Reviewer_xsGU · 2023-12-07

**Summary Of Contributions:**

This paper evaluates GPT-4 on the Abstraction and Reasoning Corpus (ARC). It finds that GPT-4 struggles on a text-based linearized representation of the tasks in the corpus, analyzes why it fails, and proposes a new version of ARC (1D-ARC) that uses 1D tasks instead of the 2D grid originally included in ARC. Finally, the paper applies ARGA (Xu et al. 2023) to compute graph-based representations of the task instances on which they evaluate GPT-4's reasoning, and find that with this representation of the task GPT-4 performs better.

**Audience:**

Yes

**Claims And Evidence:**

Yes

**Requested Changes:**

Summary of weaknesses to address:
* Some of the framing around the capabilities of LLMs here is not completely accurate. Providing GPT-4 a more usable representation of the task as computed by ARGA doesn't mean GPT-4 is solving the task; it means a broader system is proposed that improves performance on the task.
* Clearer motivation of the evaluation
* There should be more details about how the analysis was performed in Sections 2.4 and 4.3.
* Evaluation on more models

Questions:
* In the results, there are 50 tasks. What does it mean to "solve" a task? Are there multiple problem instances per task, and "solving" one means successfully predicting the 6th frame for every instance of the task? Or is there only one instance per task?
* Shouldn't the goal of selecting 50 examples to evaluate serve as a _lower_ bound rather than an upper bound on ARC as a whole?
* I realize that this paper was only recently published, but Mirchandani et al. 2023 also perform evaluation on ARC. How does this evaluation compare to the one proposed in this paper?
* How might tokenization act as a confounder even in the 1D case? E.g., did you evaluate with multiple kinds of pixel representations (entire words, numbers, random tokens as used in Mirchandani et al. 2023) for the 1D case and see how robust performance is over them?
* Why is Table 4 showing factor improvements rather than absolute improvements? It seems much less meaningful when the numbers (and dataset size in general) are already really small

Minor points:
* Evaluation on "GPT" is ambiguous as there are tons of different models fitting under the general category of "GPT". These instances should be replaced with GPT-4

**Strengths And Weaknesses:**

Strengths:
* The 1D-ARC corpus is a good contribution. I am curious how it differs from existing string manipulation tasks, though, like the dataset introduced by Andreas et al. 2018 (and used in more recent work, e.g., Grand et al. 2023).
* Investigating the representation of task instances that can be used as input to a language-only model is interesting.
* The analysis of failures, e.g., in Section 2.4, seems interesting (though I would like to see a lot more detail about how this analysis was performed).
* The experiment with rotation is interesting

Weaknesses:
* I disagree with some of the framing: when we cater to an LLM by providing it a better input representation, is it that the LLM is now able to complete the task and solve the corpus, or have we just constructed some broader system that can do it?
* I struggle to understand some of the background motivation on evaluating LLMs on a task like ARC which is fundamentally a visual reasoning task. Specifically, the final sentence of the paper: "the necessity for substantial advancements to enable reliable reasoning by LLMs". Why do we think in particular that solving ARC is necessary for building "reasoning" LLMs?
* Evaluation is only performed on GPT-4 and 3.5. Why not evaluate on the many other publicly available language models?

---

> ### Author Response · Authors · 2023-12-17
>
> Thanks for your thorough review and questions.
>
> **Framing**
>
> We appreciate your concerns regarding the true capability of LLMs in solving tasks when external tools are employed as aids. We agree that the agent responsible for solving is not solely the LLM itself, but rather the entire system. Augmented LLMs are a very active area of study and our goal is to study the reasoning capability of such systems. We will further clarify the framing in the revision.
>
> **Motivation**
>
> We recognize your concern regarding the evaluation of LLMs on the ARC, which is primarily seen as a visual reasoning task. However, it's important to note that the solutions to ARC problems are significantly language-based. Research has demonstrated that about 80% of ARC tasks can be solved by humans who subsequently provide logical, language-based explanations for their solutions [1]. Additionally, state-of-the-art ARC solvers predominantly employ domain-specific languages, indicating the critical role of linguistic processing in these solutions.
>
> This alignment with language-based reasoning underlines the appropriateness of utilizing LLMs for the ARC. The fundamental knowledge priors required for solving ARC tasks are intrinsic to human reasoning [2]. Therefore, if an LLM is to emulate human-like reasoning capabilities, its ability to process and apply these core knowledge priors is essential. Hence, we argue that successfully solving ARC tasks is a significant step towards developing LLMs capable of reliable reasoning, mirroring the cognitive processes humans use in such tasks.
>
> **Further Evaluation with other LLMs**
>
> As per the global response, we are currently experimenting with Llama 2 and will report these findings in an upcoming response as well as in the paper revision.
>
> **Details for analysis**
>
> To make our paper more readable, we moved the details of our analysis to technical appendices. If the reviewer finds that any specific details are missing, please let us know and we will be happy to add them.
>
> **Comparison with other string manipulation tasks**
>
> 1D-ARC was designed to inherit the same core knowledge priors assumed by the ARC. These priors aim to emulate the same knowledge priors inherited by humans which differs from most existing string manipulation tasks [2].
>
> We apologize but we were unable to identify the datasets referenced by the reviewer (Andreas et al. 2018, Grand et al. 2023). If you could kindly provide the title for the papers referenced, we will be able to provide a more comprehensive comparison.
>
> **Questions**
>
> *What does it mean to "solve" a task?*
>
> Each ARC task comprises 2-5 training instances and a single test instance. We consider a task to be 'successfully solved' when the system accurately predicts the exact output for the test instance.
>
> *Upper bound on ARC*
>
> Since the 50 examples selected are tasks we deemed the easiest out of the entire ARC, the performance of LLMs on these tasks is better than on the entire ARC. Therefore, the results obtained from this subset serve as an upper bound for potential performance across the entire ARC dataset.
>
> *Compare to Mirchandani et al.*
>
> Mirchandani et al’s methodology is similar to our direct-grid approach. Their findings echo ours in that LLMs fall short of the state-of-the-art models. They evaluated the complete ARC dataset which consists of 400 training tasks and 400 evaluation tasks, and found an accuracy of 85/800 for the best LLM model (table 1 in [3]), whereas our direct-grid approach yielded 81/400 on training tasks without assessing the 400 evaluation tasks. This difference likely stems from the much higher difficulty level of the evaluation tasks.
>
> *Tokenization for the 1D case?*
>
> We conducted extensive experiments to study the performance of different variations of prompting, including a variety of tokenizations on the 50-task subset.
>
> We did not extend this level of detailed experimentation to other datasets due to cost constraints. However, by rigorously evaluating and selecting the best-performing variation for these representations on the subset, we believe our findings are indicative of the likely performance across other datasets we examined, including the 1D-ARC.
>
> *Factor improvements rather than absolute improvements?*
>
> We opted for factor improvements to better illustrate the comparative efficacy of different approaches. A factor improvement, such as a 1.76x increase, conveys a clearer picture of performance enhancement than absolute numbers, like a +10 increase from 13 to 23. However, we acknowledge that in instances where both numbers are small, the significance of factor improvements can be less meaningful. We will adjust the tables to include absolute improvements as per your suggestion.
>
> [1] Johnson, Aysja, et al. Fast and flexible: Human program induction in abstract reasoning tasks.
>
> [2] Chollet, François. On the measure of intelligence.
>
> [3] Suvir, Mirchandani, et al. Large language models as general pattern machines.

---

> > ### Comment · Reviewer_xsGU · 2024-01-04
> >
> > Thank you for your response. A few further comments:
> >
> > > The fundamental knowledge priors required for solving ARC tasks are intrinsic to human reasoning [2]. Therefore, if an LLM is to emulate human-like reasoning capabilities, its ability to process and apply these core knowledge priors is essential. Hence, we argue that successfully solving ARC tasks is a significant step towards developing LLMs capable of reliable reasoning, mirroring the cognitive processes humans use in such tasks.
> >
> > Discussion of intelligence (as in the cited work) aside, I'm actually dubious of this claim -- specifically the framing of this capability as a matter of priors (that could be encoded in a static language model's parameters), rather than of underlying dynamic cognitive processes like hypothesis search and testing. I would be surprised if there is evidence from cognitive science literature showing that language is a fundamental cognitive process in solving ARC for humans (this is not in contradiction with the finding that most tasks can be explained with language). The priors that are encoded in a LLM are certainly useful, in terms of the abstractions that are most likely to be referred to in text, and so it's not totally surprising they perform well on a text representation of ARC, but when so much of what makes this work is stringifying the examples correctly, it's unclear to me whether this evaluation of ARC is probing the ability to perform on-the-fly hypothesis formation and testing that's required for something like ARC (especially when language is unavailable).
> >
> > Another way of stating this: there is evidence that the "core knowledge" as discussed in Chollet section III (e.g., objectness) is innate to human babies, who haven't yet acquired language. Not to say babies can perform ARC, but much of our own learning process when we are young and lack language competency involves the kind of on-the-fly abstraction and hypothesis formation that ARC presents.
> >
> > > We apologize but we were unable to identify the datasets referenced by the reviewer (Andreas et al. 2018, Grand et al. 2023). If you could kindly provide the title for the papers referenced, we will be able to provide a more comprehensive comparison.
> >
> > https://aclanthology.org/N18-1197/
> > https://arxiv.org/abs/2310.19791
> >
> > > Since the 50 examples selected are tasks we deemed the easiest out of the entire ARC, the performance of LLMs on these tasks is better than on the entire ARC.
> >
> > I see, this makes sense.

---

> > > ### Author Response · Authors · 2024-01-11
> > >
> > > Thank you for your further comments.
> > >
> > > > I would be surprised if there is evidence from cognitive science literature showing that language is a fundamental cognitive process in solving ARC for humans
> > >
> > > We agree with the reviewer's concerns on this point and acknowledge that we may have overstepped in our previous rebuttal claims. However, it's important to note that we did not make any claims in the paper that language is a fundamental process innate in humans.  Nonetheless, if the reviewer feels there are any claims in the revised paper that require further qualification or clarification, please let us know.
> > >
> > > > Comparison of 1D-ARC and Regular Expressions dataset in Andreas et al. 2018.
> > >
> > > Indeed this dataset bears some similarity to 1D-ARC. For example, consider our recolor by size task where:
> > > ```
> > > ⬛⬛⬛🟨🟨🟨⬛🟨⬛⬛🟨🟨⬛⬛
> > > becomes:
> > > ⬛⬛⬛🟥🟥🟥⬛🟧⬛⬛🟫🟫⬛⬛
> > > ```
> > > If we represent the colors using letters:
> > > ```
> > > bbbyyybybbyybb
> > > becomes:
> > > bbbrrrbobbwwbb
> > > ```
> > > This looks like a task from the dataset by Andreas et al. However, the regular expressions dataset does not use the knowledge priors assumed by the ARC and therefore was not designed to include concepts such as “objectness”. The previous example can be solved without the concept of objects: “replace string of yyy with rrr, yy with ww and y with o”. However, there are tasks that require priors such as objectness to be solved.
> > >
> > > For example, consider our dynamic movement task:
> > > ```
> > > ⬛⬛⬛🟨⬛⬛⬛🟥🟥🟥⬛⬛
> > > becomes:
> > > ⬛⬛⬛🟨🟥🟥🟥⬛⬛⬛⬛⬛
> > > ```
> > > represented in letters:
> > > ```
> > > bbbybbbrrrbb
> > > becomes:
> > > bbbyrrrbbbbb
> > > ```
> > >
> > > This task cannot be easily solved using regular expressions and without the concept of objects.
> > > In conclusion, while the dataset introduced by Andreas et al. shares some similarity with our dataset, it is different in that it does not assume the same knowledge priors and therefore does not cover the same object reasoning capabilities that we aimed to evaluate in our work.

---

### Author Response · Authors · 2023-12-17

We thank the reviewers for their insightful comments.

One of the suggestions made by multiple reviewers is the expansion of the results to evaluate beyond the GPT models. While we maintain our belief that evaluating the top-performing LLM would provide representative insights into the peak potential of current language models, we recognize the merit in examining a broader range of LLMs to gain a more comprehensive understanding of their reasoning abilities.

We are currently conducting additional experiments on Llama 2 models. We will include the new results in a subsequent response as well as revise the paper in the coming days. For now, please find below answers to your questions and comments on how we will address reviewer concerns on revision.

---

> ### Author Response · Authors · 2023-12-22
> **Paper Revision**
>
> We again thank the reviewers for their insightful comments and critiques. We've worked very hard to address the key experimentation and revision requests in the past two weeks since receiving the reviews and we believe these changes have substantially improved the paper. We have revised the paper and outline the changes as follows:
> - We conducted new experiments using the LLAMA2-13B and LLAMA2-70B models, the results of which have been incorporated into Tables 2 and 4.
> - We added clarification to highlight that it is the integrated system solving the ARC tasks, rather than the LLMs operating independently.
> - Table 4 has been updated to present absolute percentage improvements, replacing the previously used factor improvements for greater clarity.
> - We have enhanced our discussion on the experimentation with GPT-4V in the main article and included a figure to illustrate an example.
> - The related works and future work sections have been expanded.

---

> > ### Comment · Reviewer_uAxr · 2024-01-03
> > **Re: Paper Revision**
> >
> > Thank you for submitting the revised manuscript!

---

### Decision · Action_Editor_ewXk · 2024-01-24

**Recommendation:** Accept as is

**Comment:**

The reviewers were generally happy with the final version of the manuscript. I think the work makes a valuable point when it comes to evaluation on ARC, and the results are appropriately couched with respect to where the gains come from (essentially an external preprocessing system). It serves as food for thought for others trying to make claims (positive or negative) about LLMs' fundamental abilities, since these are typically conflated with the task format as in ARC.

The main weaknesses I see are this work lacking breadth of LMs and being out of date potentially soon (although that's true of any paper in this fast-moving area) and that the results might be unsurprising (uAxr). However, it clears the bar for TMLR's review criteria.

**Audience:**

Yes, this work is likely of interest to many researchers working on reasoning and understanding the frontier of capabilities of large language models.

**Claims And Evidence:**

The paper supports its main claims about GPT's performance on ARC, now bolstered with additional results from Llama 2 (addressing RCQt).

Moreover, concerns from reviewer xsGU have been addressed, with the paper qualifying that the object-based representation is an external system to the LLM itself.

Regarding xsGU's point about the tasks being visual in nature and language not necessarily being a process needed to solve them, I agree, but I think the introduction of the present draft is clear here. If we do believe that LLMs are "AGI", then they should (at least) be able to do these tasks, and this paper explores and important evaluation question there regarding the formatting of the task.